# The Ligand Binding Domain of the Cell Wall Protein SraP Modulates Macrophage Apoptosis and Inflammatory Responses in *Staphylococcus aureus* Infections

**DOI:** 10.3390/molecules30051168

**Published:** 2025-03-05

**Authors:** He Sun, Robert W. Li, Thomas T. Y. Wang, Lin Ding

**Affiliations:** 1College of Mechanical and Electrical Engineering (CMEE), Central South University, Changsha 410083, China; 2Animal Parasitic Diseases Laboratory, United States Department of Agriculture, Agricultural Research Service (USDA-ARS), Beltsville, MD 20705, USA; 3Diet, Genomics, and Immunology Laboratory, Beltsville Human Nutrition Research Center, United States Department of Agriculture, Agricultural Research Service (USDA-ARS), Beltsville, MD 20705, USA; 4Department of Core Facility of Basic Medical Sciences, Shanghai Jiao Tong University School of Medicine, Shanghai 299925, China

**Keywords:** adhesin, antibiotics, gain-of-function, loss-of-function, methicillin-resistant *Staphylococcus aureus*, SraP, virulence

## Abstract

The *Staphylococcus aureus* cell wall protein serine rich adhesin for platelets (SraP) belongs to a large surface glycoprotein family of adhesins. Here, we provide experimental evidence that SraP mediates macrophage functions in a human monocyte-derived macrophage model via its N-terminal L-lectin module (LLM) in the ligand binding region. Our flow cytometry data demonstrated that macrophages infected by the LLM deletion strain profoundly impacted apoptosis, reducing the percentage of apoptotic cells by approximately 50%, whereas LLM overexpression significantly increased the percentage of early-stage apoptotic cells (*p* < 0.001). LLM deletion significantly enhanced phagocytosis by macrophages by increasing the number of engulfed bacteria, resulting in a significant increase in bacterial killing and leading to a notable decrease in bacterial survival within macrophages (*p* < 0.001). Furthermore, LLM modulated the ability of *S. aureus* to elicit inflammatory responses. The LLM deletion strain dampened the expression of proinflammatory factors but increased the expression of anti-inflammatory cytokines, such as IL10. Our evidence suggests that SraP likely plays a dual role in *S. aureus* pathogenesis, by acting as a virulence factor involved in bacterial adhesion and invasion and by mediating macrophage functions. Our future work will focus on the identification of small molecule inhibitors of LLM using molecular docking-based in silico screening and in vivo validation. Developing LLM inhibitors, alone or in combination with conventional antibiotics, may represent a novel strategy for combating *S. aureus* infections.

## 1. Introduction

Infections by methicillin-resistant *Staphylococcus aureus* (MRSA) represent a serious threat to public health. In the United States, the number of MRSA cases is estimated to be approximately 3% in the general population with the death rate reaching to 2.7% among patients with healthcare-associated infections, according to a 2020 report by the Centers for Disease Control and Prevention [1]. A recent meta-analysis suggested that, alarmingly, the global prevalence of MRSA could be as high as 15% among the residents of elderly care centers [2]. The primary therapeutic option against MRSA infections is still antibiotics. Dozens of antibiotics have been approved to treat both community-acquired (CA) and healthcare-associated (HCA) MRSA [3,4]. Sulfamethoxazole/trimethoprim (bactrim), clindamycin, daptomycin, minocycline, vancomycin, and linezolid are among the most frequently used antibiotics. Depending on the genotypes of the pathogens as well as the severity and locations of the infection, the choice of antibiotic types and treatment duration differs substantially. The decision to choose a particular antibiotic for a given MRSA strain is often neither evidence driven nor guided by genetics. Large-scale cost-benefit analyses for specific treatment options are still lacking. Moreover, many strains of MRSA have developed resistance to even newly developed antibiotics. For example, vancomycin is an important first-line antibiotic for the treatment of MRSA [5]. Since 2002, at least 52 vancomycin resistant MRSA strains have been isolated globally [6]. The rapid spread of antibiotic-resistant MRSA has compromised the efficacy of existing drugs. While multiple new classes of compounds for treating antibiotic-resistant MRSA have been developed [7], few emerging options for MRSA treatment have been approved by the authorities or recommended by the Infectious Disease Society of America (IDSA), necessitating novel therapeutics and control strategies.

Serine-rich adhesin for platelets (SraP) is one of the key *S. aureus* proteins and able to directly bind to host cells via receptor-ligand interactions. SraP also contributes to MRSA virulence [8]. SraP belongs to the serine-rich repeat glycoprotein (SRRP) adhesin family, with an estimated molecular mass of 227 kDa. An early study suggested that SraP promotes bacterial aggregation and has dual roles as a host and bacterial adhesin [9]. A recent study revealed that the protein possesses an extended rod-like architecture with four discrete modules, the L-lectin module (LLM), a β-grasp fold, and two tandem cadherin-like modules, in this order [10].

A mutation in the full-length ligand binding region (LLM) substantially abolished the bacterial adhesion capacity to human lung epithelial cells in vitro, resulting in a 50% reduction in invasion, whereas the other three modules have no measurable effect on cell adhesion and invasion [10]. In a separate study, the LLM deletion mutant exhibited a ∼50% decrease in adhesion and invasion, compared with those of wild-type (WT) *S. aureus* [11]. The passive immunization of mice with anti-LLM monoclonal antibodies conferred protection against challenge infection with the MRSA strain USA300, resulting in a significant decrease in the blood bacterial load [11], providing proof of concept that LLM can serve as a druggable target for efficacious MRSA control.

*S. aureus* invades a broad range of host cells, including platelets, endothelial and epithelial cells, enterocytes, and professional phagocytes (e.g., macrophages and neutrophils). A number of bacterial adhesins, including clumping factors and fibronectin-binding proteins, are involved in colonization and invasion. These adhesins recognize and bind to various platelet surface proteins, such as fibrinogen and fibronectin. For example, in addition to its major ligand, clumping factor A also interacts with a 118 kDa platelet membrane protein [12]. The direct binding of SraP to platelets can be rapid, acting in a receptor-ligand manner [8]. Moreover, *S. aureus* attachment to epithelial cells represents a critical event during bacterial pathogenesis. For example, scavenger receptor class F member 1 (SRECI) recognizes and binds to the cell wall teichoic acid of *S. aureus*, directly affecting bacterial colonization [13]. After colonization and biofilm formation, the pathogen releases an array of virulence factors, subsequently activating host immune responses and regulating host physiology. SraP-mediated binding of *S. aureus* to epithelial cells likely involves its interaction with sialylated (sialic acid) receptors on host cells [10]. LLM specifically binds to the N-acetylneuraminic acid (Neu5Ac) moiety of glycoproteins, particularly sialylated receptors, on the surface of host cells [10]. This process is complex and plays dual roles in enhancing bacterial colonization and invasion and allowing immune escape. *S. aureus* interactions with macrophages are also receptor-mediated. SraP interacts with sialylated surface glycoproteins, which are abundant on macrophages. These glycoproteins belong to the sialic acid binding immunoglobulin-like lectin (SIGLEC) family. Sialoadhesin (SIGLEC1 or CD169) and SIGLEC11 are predominantly expressed on macrophages, whereas SIGLEC3 (CD33), SIGLEC5 (CD170), SIGLEC9 (CD329), and SIGLEC10 are common on monocytes and other immune cells [14]. These SIGLECs have various degrees of preference for Neu5Ac-containing moieties, suggesting that they may play direct roles in SraP-mediated pathogen-macrophage interactions [15]. In addition, macrophage receptors with collagenous structure (MARCO), a human class A scavenger receptor, also serves as a major receptor for bacterial binding to alveolar macrophages [16]. Although both tissue-resident and monocyte-derived macrophages can effectively engulf and eliminate *S. aureus* cells, a small but significant number of engulfed *S. aureus* can become resistant to bactericidal attack inside vacuolar compartments and survive, contributing to persistent infections [17]. While its role in bacterial adhesion and invasion is well recognized, little is known about whether the intact SraP or its LLM possesses any other biological properties, such as phagocytosis and subsequent killing by macrophages. In this study, we designed in vitro loss-of-function (LOF) and gain-of-function (GOF) experiments [18] to understand the potential roles of LLM in host-pathogen interactions and in mediating macrophage functions using human monocyte-derived macrophages as a model.

## 2. Results

### 2.1. The L-Lectin Module Affects Macrophage Apoptosis Only in Its Early Stage

To gain insight into how LLM may affect the function of macrophages, we generated *S. aureus* mutant strains with LLM deletion (LOF) or overexpression (GOF). Our PCR results revealed that LLM was absent in the LLM deletion (ΔL-lectin) mutant strain, whereas its expression was >2000-fold greater in the LLM-overexpressing (GOF) strains than in their WT counterparts as demonstrated by sequencing and quantitative RT PCR (Appendix A). These two strains were subsequently used to infect macrophages, and their effects on apoptosis at different stages were determined. Cells undergoing early-stage apoptosis can be stained specifically with fluorescein isothiocyanate (FITC)-labeled annexin V, which binds with high affinity to negatively charged phosphatidylserine (PS) in the presence of Ca^++^. An increased expression of PS on the outer membrane is one of the hallmarks of early-stage apoptotic cells. As a result, it serves as an excellent target for noninvasive imaging of apoptosis [19]. On the other hand, late-stage apoptotic cells can be distinguished via the use of propidium iodide (PI), a nucleic acid dye that penetrates the cell membrane and stains the nucleus red. As Figure 1A shows, cells in the Q3 quadrant region [(Annexin V-FITC)^+^/PI^−^] represent the early stage of apoptosis, while cells in the Q2 region [(Annexin V-FITC)^+^/PI^+^] are at the late apoptotic stage and live cells are both annexin V- and PI- negative. Figure 1B quantifies that LLM deletion (ΔL-lectin) significantly decreased the percentage of early apoptotic cells in the population, from 6.00 ± 1.49 to 1.89 ± 0.18 (mean ± SD, *p* < 0.01). In contrast, the proportion of early apoptotic macrophages after infection with the LLM-overexpressing GOF strain (WT pRMC L-lectin) significantly increased from 3.89 ± 0.27 to 9.07 ± 0.29 (Figure 1B, *p* < 0.001), which was substantially greater than that in the macrophage population infected with the WT MW2 strain. However, the impact of LLM on apoptosis at the late stage was subdued, and the percentage of late apoptotic cells in the LOF group (ΔL-lectin) was only marginally lower than that in the group infected with the WT strain (*p* > 0.05). Conversely, the percentage of late apoptotic cells in the LLM GOF group was nominally greater than that in the empty vector (pRMC2) group (Figure 1C, *p* > 0.05).

The expression of apoptosis-related genes at the mRNA level was examined using quantitative RT-PCR. Our data revealed that the expression levels of the proapoptotic factors BCL2 associated with X, apoptosis regulator (*BAX*), and Fas cell surface death receptor (*FAS*) in the ΔL-lectin group were 2.55-fold (*p* < 0.001) and 1.87-fold (*p* < 0.001) lower than those in the group infected with the WT strain, respectively. On the other hand, the expression levels of these two genes were 2.43-fold (*p* < 0.05) and 2.24-fold (*p* < 0.05) greater in the group infected with the strain overexpressing LLM than in the vector control group, respectively. Conversely, the mRNA expression level of the antiapoptotic factor BCL2 apoptosis regulator (*BCL2*) was 1.70-fold greater in the ΔL-lectin group than in the WT group (Figure 2C, *p* < 0.01), whereas its expression level was reduced by 2.51-fold in the LLM overexpression group (Figure 2C, *p* < 0.01).

### 2.2. The SraP L-Lectin Module Plays a Critical Role in Phagocytosis and Bacterial Killing by Macrophages

To investigate whether LLM affects phagocytosis of *S. aureus* and its survival within macrophages, we infected THP-1-derived macrophages with the WT strain (MW2), the L-lectin deletion strain, WT with the empty vector *pRMC2*, or the *pRMC* LLM overexpression strain. We then measured the phagocytic uptake of different strains by THP-1-derived macrophages and assessed the survival of these strains engulfed by macrophages. The results revealed that the number of bacteria phagocytosed by macrophages increased from 9.62 (±1.86) × 10^10^ CFU/mL to 1.32 (±0.94) × 10^11^ CFU/mL in the ΔL-lectin group compared with the WT group (*n* = 4, *p* < 0.01; Figure 3A). However, the number of phagocytosed bacteria was unchanged in the GOF (overexpression) group compared with the group infected with the WT containing the empty expression vector (*p* > 0.05). Furthermore, LLM significantly affected the survival of bacteria engulfed by macrophages. After LLM deletion, the number of surviving bacteria inside the macrophages decreased by ~50%, from 2.13 (±2.05) × 10^11^ CFU/mL to 1.09 (±1.63) × 10^11^ CFU/mL in the WT group (*n* = 4, *p* < 0.001; Figure 3B). Nevertheless, LLM overexpression had no significant effect on bacterial survival.

To elucidate the possible underlying mechanism responsible for the increased phagocytosis of *S. aureus* following LLM deletion, we quantified the expression levels of Fcγ receptors and complement receptors (CRs) in macrophages, which play a vital role in initiating phagocytosis by binding to opsonized bacteria, infected with either the ΔL-lectin strain or the WT strain. Our data revealed that the mRNA levels of several phagocytosis-related genes, including the receptors FcγRI (CD64) and FcγRIII (CD16), in the group infected with ΔL-lectin were 3.22-fold (Figure 4A, *p* < 0.001) and 2.43-fold (Figure 4B, *p* < 0.001) greater than those in the WT group, respectively. Furthermore, the mRNA level of CR3, a complement receptor involved in pathogen pattern recognition, was also significantly greater in the ΔL-lectin group than that in the WT group (Figure 4C, *p* < 0.001).

### 2.3. The L-Lectin Module Modulates the Ability of Staphylococcus aureus to Elicit Inflammatory Responses in Macrophages

Compared with those infected with the WT strain, macrophages infected with the ΔL-lectin strain presented more rapid restoration of the pro- and anti-inflammatory balance. Compared with those in the WT infection group, the expression levels of common proinflammatory factors in the ΔL-lectin group were significantly lower, including IFNγ, IL1β, IL6, IL8, and TNFα (Figure 5A–E). On the other hand, the expression levels of anti-inflammatory factors, such as ARG1 (*p* < 0.05, Figure 5F), IL10 (Figure 5G), and TGFβ (Figure 5H), were significantly greater in the group infected with the ΔL-lectin strain than in the WT group. We also assessed the expression levels of several key transcription factors in relevant signaling pathways, including the NF-κB, STAT1, and PI3K/Akt signaling pathways. Compared with infection with the WT strain, infection with the LLM deletion strain significantly reduced the mRNA expression of nuclear factor kappa B subunit 1 (*NFKB1*) (3.37-fold, *p* < 0.001; Figure 6A) and AP1 or JUN (3.42-fold, *p* < 0.001; Figure 6B) in the NF-κB and MAPK/AP-1pro-inflammatory signaling pathways. Moreover, NLR family pyrin domain containing 3 (*NLRP3*), an intracellular sensor capable of detecting a broad spectrum of microbial motifs [20], was more readily induced by the WT strain than by the LLM deletion strain (*p* < 0.05, Figure 6C). Nevertheless, LLM did not appear to have any significant effect on the expression of other key transcription factors examined, including *STAT1* and *JAK1* (Appendix A).

## 3. Discussion

SraP is a member of a large surface glycoprotein family of adhesins and is structurally similar to the previously characterized GspB [9,21]. SraP is widely expressed in many clinical MRSA isolates. For example, the gene encoding SraP is present in all six sequenced MRSA genomes [8]. The crystal structure of the ligand-binding region of SraP has been characterized [10], which consists of four discrete modules, including the N-terminal LLM and three other modules, providing structural insights into its role in host-pathogen interactions. LLM is solely responsible for the specific binding of *S. aureus* to human epithelial cells, contributing to its virulence. Mutations in either the full-length SraP ligand-binding region or its LLM abolish the ability of MRSA to adhere to host cells. For example, the deletion of SraP leads to an approximately 40% decrease in adhesion and results in a ~50% decrease in the level of invasion, compared with WT [10]. Aside from its roles in adhesion and invasion, other biological functions of SraP have yet to be elucidated.

In this study, for the first time, we provide experimental evidence that SraP plays an important role in mediating the function of human monocyte derived macrophages, including their abilities related to apoptosis, phagocytosis, and bacterial killing. Our data revealed that the number of bacteria phagocytosed by macrophages significantly increased due to LLM deletion. However, the number of cells that survived after their engulfment in macrophages decreased by approximately 50%, thanks to the LLM deletion. LLM also markedly affected apoptosis, particularly during its early stage, as well as the ability of *S. aureus* to elicit proinflammatory responses. Our findings suggest that SraP likely plays a dual role in MRSA pathogenesis. In addition to its direct role in bacterial adhesion and invasion, SraP can compromise macrophage-directed bacterial killing and, subsequently, host immunity.

*S. aureus* interacts with macrophages via multiple biological processes, including receptor-mediated endocytosis and phagocytosis [22]. The latter represents the primary pathway used by macrophages to engulf *S. aureus*. Multiple macrophage receptors are involved in this process. Scavenger receptors (such as MARCO) bind directly to *S. aureus* [16], whereas Fc receptors and complement receptors indirectly interact with *S. aureus* [23]. SraP is also involved in *S. aureus*-macrophage interactions via sialic acid (sialylated) receptor-mediated processes. Our data show that several sialic acid receptors, including SIGLEC1 and SIGLEC11, are expressed in human monocyte-derived macrophages, as previously described [17]. Both receptors are responsive to bacterial infections as well as bacterial cell wall components, such as lipopolysaccharides (LPS) and peptidoglycans (unpublished). LLM specifically recognizes the Neu5AC moiety of SIGLEC1 expressed on macrophages. In addition to macrophages, many other immune cells express specific SIGLECs. For example, CD22 (SIGLEC2) is expressed on B cells and binds to the tumor prognosis marker Neu5AC-α2–6-GalNAc [24]. The mutant form of CD22 is unable to bind to sialylated glycans, compromizing B cell signaling and survival [25]. SIGLEC7, primarily expressed on natural killer cells and monocytes, also possess sialic acid binding properties. It is conceivable that SraP can affect cellular functions in a wide range of immune cell types.

Our findings revealed that LLM deletion significantly upregulated the expression of both complement receptor 3 (CR3), a heterodimer of the CD11b (α) and CD18 (β) transmembrane glycoproteins, and Fc receptors (Figure 3). The activation of the complement cascade by binding complement factors C3b and iC3b to CR3 (and/or CR4) promotes the internalization of opsonized *S. aureus*. These complement proteins also accelerate the recruitment of phagocytes (e.g., macrophages) to the infection site. Furthermore, *S. aureus* can be directly targeted by the formation of a membrane attack complex to cause bacterial lysis, possibly via multiple pathways, such as the release of reactive oxygen species (ROS), lysozyme-directed killing, and acidification. A low phagosomal pH environment negatively affects the survival of *S. aureus*. The effect of LLM on macrophage apoptosis may also contribute to the survival of *S. aureus* pathogens. Moreover, the interaction between Fc receptors on the macrophage surface and the Fc region of antibodies enhances the opsonization of invading *S. aureus* pathogens, enabling rapid engulfment. Nevertheless, our knowledge of either the entire SraP or its LLM is still in its infancy. Additional in vivo experimental evidence is needed to enhance our understanding of the biological function of SraP. For example, how adhesins and various macrophage receptors act simultaneously to coordinate the engulfment of invading *S. aureus* warrants further investigation.

We speculate that chemical inhibition of SraP or LLM may have multiple advantages over conventional approaches for the development of novel antibiotics, including those targeting quorum sensing [26]. First, targeting LLM may disrupt MRSA colonization in its early stage. LLM inhibition also presents a valuable opportunity to modify host-pathogen interactions, including enhancing macrophage functions in the host. It is conceivable that developing LLM inhibitors, alone or as part of combination therapies with traditional antibiotics, may represent a novel yet logical strategy for combating MRSA infections. The ability of the SraP mutant strain to initiate infection is significantly impaired compared with that of its parent strain [8], suggesting that SraP is involved in *S. aureus* virulence. Moreover, SraP is expressed in approximately 85% of clinical MRSA isolates, providing further evidence that its role in virulence is broadly based. *S. aureus* is known to secrete a diverse array of virulence factors, including microbial surface components recognizing adhesive matrix molecules (MSCRAMMs), such as various adhesins and biofilm-associated proteins [27]. MSCRAMMs bind to various proteins on host cells, playing a critical role during the early stage (colonization) of MRSA pathogenesis. For example, fibronectin-binding proteins A and B are responsible for bacterial attachment to fibronectin, fibrinogen, and elastin expressed on host cells [28]. As one of the MSCRAMMs, SraP is able to recognize host receptors. As a result, targeting SraP may represent a novel anti-virulence approach to fight MRSA infection. In addition to antibodies, high-throughput screening for small molecules and/or natural products that are able to disrupt the interactions between SraP and host receptors is highly desirable. Moreover, while the in vitro evidence provided in this study is of interest, developing in vivo prophylactic and curative models appears to be a high priority. Notably, SraP is just one of multiple factors implicated in bacterial adhesion to host cells. Special attention should be focused on developing strategies simultaneously targeting multiple factors secreted during the course of MRSA colonization.

## 4. Materials and Methods

### 4.1. Bacterial Culture Conditions

The MRSA strain MW2 was obtained from Biosea (Beijing, China) and grown in Mueller Hinton (MH) broth. Briefly, frozen stocks of the primary culture were streaked out on agar plates. A single colony was used to inoculate tryptic soy broth (TSB) media, followed by incubation in 10 mL of MH broth at 37 °C with shaking at 200 rpm for 4 h. The identity of MW2 was verified using Sanger sequencing.

### 4.2. Construction of SraP L-Lectin Module Deletion Mutants (ΔL-Lectin)

The LLM deletion (LOF) mutant was constructed via the allelic replacement technique [29]. The plasmid pKOR1, an *Escherichia coli*/*S. aureus* shuttle vector, was used to clone the amplicon flanking the SraP LLM locus. The primers used can be found in Appendix A. The primer pairs L-lectin-up-F/L-lectin-up-R and L-lectin-down-F/L-lectin-down-R were used to generate the left and right SraP LLM DNA fragments, respectively. PCR reactions were prepared as follows: 2 μL of cDNA (100 ng), 2 μL of each primer (forward and reverse, 20 nM each), 25 μL of 2× pfu PCR mix, and 19 μL of nuclease-free water. The amplification parameters included initial denaturation at 95 °C for 5 min and then denaturation at 94 °C for 30 s, 50 °C for 30 s, and 72 °C for 40 s for 2 cycles, followed by 30 cycles at 94 °C for 30 s, 55 °C for 30 s, and 72 °C for 40 s. The left and right fragments were then ligated via the primers L-lectin-up-F and L-lectin-down-R. The ligated fragments were then inserted into the pUX-T vector and transformed into *E. coli* competent DH5α cells. The fidelity of amplification and cloning was verified by resequencing. The amplification of L-lectin-UD-pUX-T and pKOR1 plasmids was performed via the primer pairs L-lectin-pKOR1-F/L-lectin-pKOR1-R and pKOR1-L-lectin-F/pKOR1-L-lectin-R, respectively. To construct the LLM-deficient plasmid pKOR1 (ΔL-lectin), the following procedures were performed. First, 2 μL of 5× infusion mixture, 2 μL of pKOR1 PCR products, and 6 μL of L-lectin PCR products were combined and transformed into *E. coli* DH5α, followed by electroporation into the MW2 strain. Subsequently, 5–10 μL of the pKOR1-ΔL-lectin-positive MW2 culture was transferred to prewarmed TSB media supplemented with 10 μg/mL chloramphenicol and grown overnight at 42 °C with vigorous shaking. The culture mixture was then streaked onto tryptic soy agar (TSA) plates supplemented with 10 μg/mL chloramphenicol, prewarmed at 42 °C, and incubated overnight. A single colony was then inoculated into 5 mL of TSB without any antibiotics and incubated at 30 °C overnight or until growth was evident. The culture mixture was then diluted 10,000-fold with sterile water, and 10–100 μL was spread onto TSA plates containing 100 ng/mL anhydrotetracycline and incubated at 37 °C. Ten large colonies were inoculated into 5 mL of TSB and grown overnight at 37 °C with shaking. DNA from the SraP LLM deletion mutant (ΔL-lectin) was purified via a miniprep kit (Qiagen, Germantown, MD, USA) and verified by PCR using the primer pair L-lectin-JD-F and L-lectin-JD-R. The identity of the LLM deletion mutant from the ten randomly picked colonies was verified using the traditional Sanger sequencing.

### 4.3. Construction of the LLM Overexpression Strain

The expression vector pRMC2 was used to construct LLM overexpression strains [30]. Briefly, the start codon AUG was added immediately prior to the LLM coding sequence while the codon UAA was inserted to serve as a stop codon. A GenScript calculator [31] was used to design the ribosome binding site (RBS) of the LLM sequence and the spacer sequence between the RBS and the start codon to improve translation efficiency. The translation rate values ranged from 1 to 100,000, with higher values indicating a higher translation initiation efficiency. The following parameters, total ribosome binding free energy (ΔG total), mRNA and 16S rRNA binding free energy (ΔG mRNA-rRNA), spacing region extension or compression free energy (ΔG spacing), and unfolding mRNA free energy (ΔG stacking). Lower values of these ΔG parameters suggest higher translation initiation efficiency. Finally, a strong bacterial transcription terminator, pBP-BBa_B0015 [32] was inserted immediately after the LLM coding sequence to construct the pRMCL-lectin overexpression plasmid (Appendix A). The constructed plasmid was heat-shock transformed into *E. coli* DH5α competent cells, followed by electroporation into the *S. aureus* restriction-deficient strain RN4220 for plasmid modification, and electroporation into the wild-type MW2 strain. Positive clones were picked and cultured in TSB containing chloramphenicol (15 μg/mL). DNA from the pRMC L-lectin overexpression strain was verified via PCR using the primer pair 5′-CACAGATGCGTAAGGAGA-3′ (F) and 5′-ATATCATTGATAGAGTTATT-3′ (R) and further validated by Sanger sequencing. The strain was induced with 500 ng/mL anhydrotetracycline for 30 min to achieve optimal LLM overexpression as a GOF model. Compared to the strain containing the empty vector, the LLM overexpression strain had greater than 2000-fold higher LLM mRNA levels, as validated using quantitative RT PCR (Appendix A).

### 4.4. THP-1 Cell Culture and Differentiation

The human THP-1 cells used in this study were obtained from MeisenCTCC (Hangzhou, China) and cultured as previously described [31]. Briefly, the cells were cultured in RPMI 1640 media containing 10% FBS, 1% penicillin-streptomycin, and 0.05 mM β-mercaptoethanol. The cells were then seeded onto a six-well plate at a density of 1 × 10^6^ cells/mL, and the medium was changed daily. The THP-1 cells were then differentiated as a model for human macrophages via phorbol 12-myristate 13-acetate (PMA) as previously described [33].

### 4.5. Flow Cytometry

THP-1 derived macrophages were cocultured with the WT strain MW2, the ΔL-lectin strain, the WT strain with pRMC empty vector, or the pRMC LLM overexpression strain in the MW2 background, at a multiplicity of infection (MOI) of 10:1 for 3 h. The cells were gently digested with 300 μL of trypsin without EDTA for 3 min and then washed with 1 mL PBS. The cells were subsequently centrifuged at 1800 rpm for 5 min. The supernatant was discarded, and the cells were gently washed again with 1 mL of PBS. The cell density was adjusted to 1.5 × 10^5^/mL. The cells were stained with 10 μL of trypan blue dye and then counted using a hemocytometer. After being washed and spun, the cells were resuspended in 100 μL of 1× binding buffer. The staining solutions of FITC (green)-labeled annexin V and propidium iodide (PI, red; 5 μL each) were used to coincubate the cells in the dark at room temperature for 10 min. The strained cells were then gently mixed in 400 μL of 1× binding buffer and analyzed via a CytoFLEX flow cytometer (Beckman, Brea, CA, USA) within 1 h.

### 4.6. Macrophage Phagocytosis and Survival Assays

A Gentamicin protection assay was used to measure phagocytosis and bacterial killing and performed as previously described [33,34]. The macrophages were infected with either the MW2 WT strain or the ΔL-lectin strain at an MOI of 20:1. After 2 h of coincubation, the bacterial suspension was aspirated from the wells, and the cells were then washed twice with PBS. Fresh media containing 50 μg/mL gentamicin was added to kill the extracellular bacteria, and the cells were then incubated for 2 h. The supernatant was discarded, and the cells were washed twice with PBS. One milliliter of precooled cell lysis buffer (10 mM EDTA, 0.25% Triton-X 100, and PBS) was then added to the cells, which were subsequently incubated on ice for 10 min. The cells were lysed completely by pipetting them up and down. After cell lysis and serial dilutions, 100 µL of the appropriate dilutions of the lysate were plated on LB agar plates. Four technical replicates were used per data/time point. After overnight incubation at 37 °C, colonies on each plate were counted manually. The mean colony number of all four plates was used to represent the number of phagocytosed bacteria (T1).

The macrophages were infected with the *S. aureus* WT MW2 strain at an MOI of 20:1. After 2 h of coincubation, the suspension was aspirated, and the cells were washed twice with PBS. Fresh complete medium containing 50 μg/mL gentamicin was then added to kill the extracellular bacteria, and the cells were incubated for either 2 or 10 h. The supernatant was removed, and the wells were washed twice with PBS. Subsequently, 1 mL of precooled cell lysis buffer was added, and the cells were incubated on ice for 10 min. The cell lysate underwent serial dilution, and 100 µL of the appropriate dilutions of the lysate were plated on LB agar plates as T1. The colonies on each of the four replicate plates were manually counted as the number of surviving bacteria, referred to as T2. The survival rate of the engulfed bacteria was expressed as the ratio of T2 to T1.

### 4.7. Gene Expression Analysis Using Quantitative RT-PCR

The total RNA was extracted from THP-1-derived macrophages as previously described [35], via the use of TRIzol reagent according to the manufacturer’s instructions. The crude total RNA was further purified via a Qiagen RNeasy Micro Kit (Qiagen, Germantown, MD, USA) with DNase I digestion. RNA integrity was verified using a BioAnalyzer 2000 instrument (Agilent, Santa Clara, CA, USA). All total RNA samples were in good quality with the RNA Integrity Number (RIN) > 8.0. An iScript advanced cDNA synthesis kit was used for cDNA synthesis from total RNA (Bio-Rad, Hercules, CA, USA). Quantitative real-time PCRs were carried out using a CFX Connect Real-Time PCR Detection System (Bio-Rad). The reactions were run in duplicate in a total volume of 22 μL containing the following: 2 μL of cDNA (100 ng), 0.5 μL of each primer (forward and reverse, 20 nM each, Appendix A), 11 μL of SsoAdvanced Universal SYBR Green Supermix, and 8 μL of nuclease-free water. Primer sequences can be found in Appendix A. The amplification reactions were subjected to initial denaturation at 95 °C for 3 min, followed by 40 cycles of 95 °C for 30 s, 60 °C for 30 s, and 72 °C for 30 s. The relative expression level of the target gene was determined via the 2^−ΔΔCT^ method [36], with GAPDH used as the housekeeping reference gene.

### 4.8. Statistical Analysis

Statistical analysis was conducted using a two-tailed *t* test implemented via SPSS 21.0 software (SPSS, Chicago, IL, USA), with a significance level of α = 0.05. The data are expressed as the means ± SDs unless otherwise stated. A *p* value ≤ 0.05 was considered statistically significant.

## Figures and Tables

**Figure 1 molecules-30-01168-f001:**
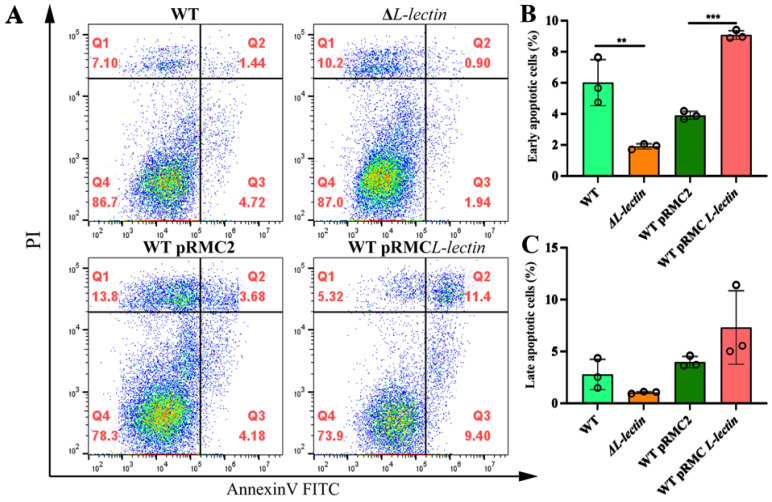
The L-lectin module (LLM) of the *Staphylococcus aureus* protein serine-rich adhesin for platelets (SraP) affects the early stage of apoptosis in macrophages infected with various MRSA strains. (**A**) The apoptosis assay was performed via concurrent staining with fluorescein isothiocyanate (FITC)-labeled annexin V and propidium iodide (PI). The cells were classified as early apoptotic with annexin V-only positive staining (the lower right quadrants of each panel or Q3), whereas late apoptotic (or dead) cells were classified as both annexin V- and PI-positive (the upper right quadrants of each panel or Q2). Live cells were both annexin V- and PI-negative (Q4). Q1 represents nonviable/necrotic cells. (**B**) The percentage of early apoptotic cells affected by LLM deletion or overexpression. (**C**) The percentage of late-stage apoptotic cells did not show ant statistical difference between the groups (*p* > 0.05). WT: wild-type MW2 strain. ΔL-lectin: LLM deletion strain. WT pRMC2: WT with empty expression vector. WT pRMC L-lectin: LLM-overexpressing strain in the WT background. The number denotes the mean and SD of three replicates. *** *p* < 0.001; ** *p* < 0.01.

**Figure 2 molecules-30-01168-f002:**
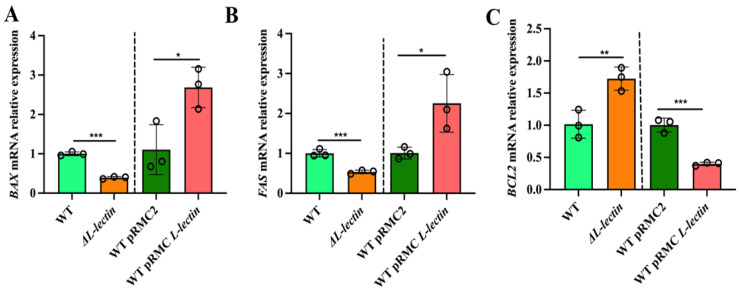
The expression of key apoptosis-related genes in macrophages infected with LLM deletion or overexpression strains was detected by quantitative RT-PCR. (**A**) *BAX* (BCL2-associated X, apoptosis regulator). (**B**) *FAS* (Fas cell surface death receptor). (**C**) *BCL2* (BCL2 Apoptosis Regulator). WT: wild-type MW2 strain. ΔL-lectin: LLM deletion strain. WT pRMC2: WT with empty expression vector. WT pRMC L-lectin: LLM overexpression strain in the WT background. The number denotes the mean and SD of four replicates. *** *p* < 0.001; ** *p* < 0.01; * *p* < 0.05.

**Figure 3 molecules-30-01168-f003:**
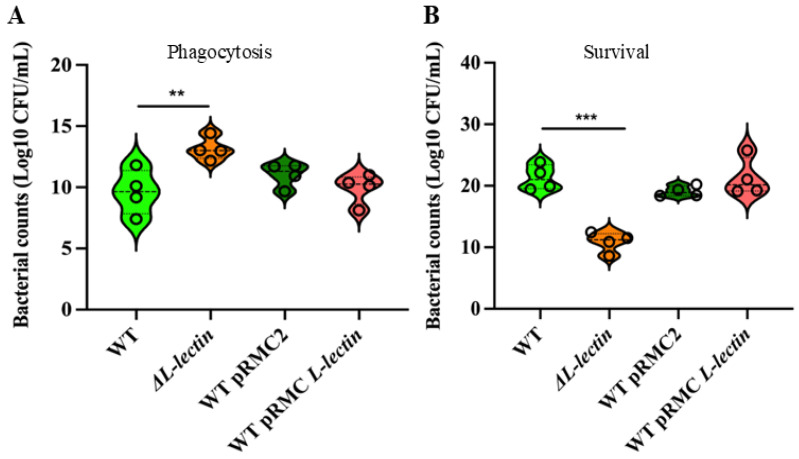
The effects of deletion and overexpression of the L-lectin module on phagocytosis and bacterial killing by macrophages were determined using a Gentamicin protection assay. (**A**) Phagocytosis is expressed as the number of bacterial cells engulfed by macrophages within a given time, whereas (**B**) bacterial killing is expressed as the number of surviving bacteria (protected from gentamicin due to intracellular localization) per time unit, as counted on agar plates. CFU: colony-forming units. WT: wild type MW2 strain. ΔL-lectin: LLM deletion strain. WT pRMC2: WT with empty expression vector. WT pRMC L-lectin: LLM overexpression strain in the WT background. The number denotes the mean and SD of four replicates. *** *p* < 0.001; ** *p* < 0.01.

**Figure 4 molecules-30-01168-f004:**
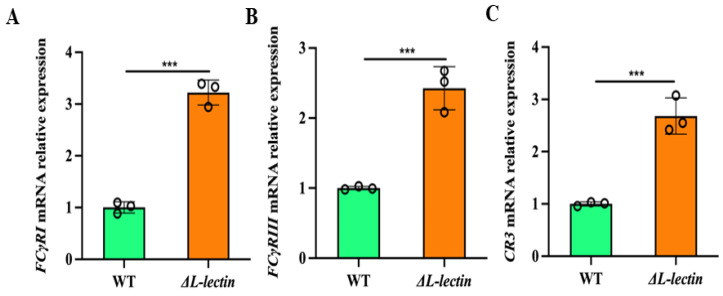
The expression of phagocytosis related genes in macrophages was affected by infection with the wild-type MRSA strain and its LLM deletion mutant. (**A**) FCγ receptor Ia (*FCGR1A*). (**B**) FCγ receptor IIIa (*FCGR3A*). (**C**) Complement receptor 3 (CR3). WT: wild-type MW2 strain. ΔL-lectin: LLM deletion strain in the MW2 background. The number denotes the mean and SD of three replicates. *** *p* < 0.001.

**Figure 5 molecules-30-01168-f005:**
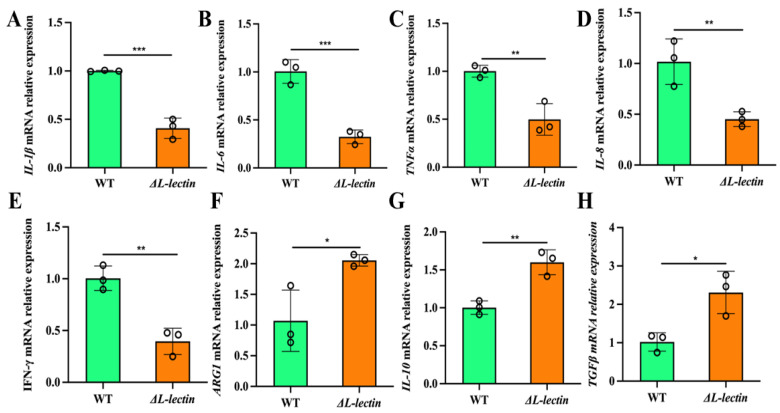
L-lectin module deletion had a profound effect on the expression of pro- and anti-inflammatory genes. (**A**) *IL-1B* (IL-1β). (**B**) *IL-6*. (**C**) *TNF* (tumor necrosis factor or TNFα). (**D**) *IL-8 (CXCL8 or C-X-C motif chemokine ligand 8)*. (**E**) *IFNG* (Interferon IFN, gamma). (**F**) *ARG1* (Arginase 1). (**G**) *IL10*. (**H**) *TGFB1* (transforming growth factor beta 1). WT: wild-type MW2 strain. ΔL-lectin: LLM deletion strain in the MW2 background. The number denotes the mean and SD of three replicates. *** *p* < 0.001; ** *p* < 0.01; * *p* < 0.05.

**Figure 6 molecules-30-01168-f006:**
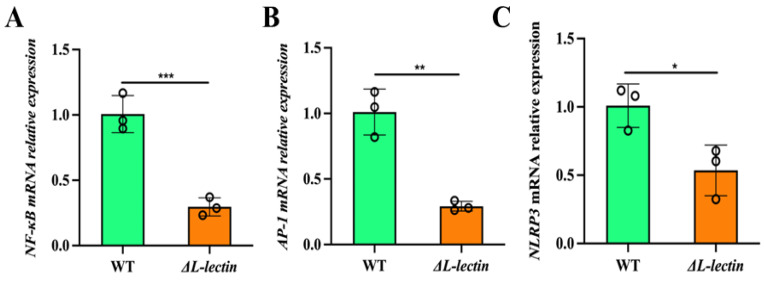
The L-lectin module loss of function downregulated the expression of key transcription factors. (**A**) Nuclear Factor Kappa B Subunit 1 (*NFKB1*). (**B**) AP-1 transcription factor (*JUN*). (**C**) NLR family pyrin domain containing 3 (*NLRP3*). WT: wild-type MW2 strain. ΔL-lectin: LLM deletion strain in the MW2 background. The number denotes the mean and SD of three replicates. *** *p* < 0.001; ** *p* < 0.01; * *p* < 0.05.

## Data Availability

All supporting materials can be found in the online Appendix A and are available from the corresponding authors upon request.

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
