# Peer review of "The Ligand Binding Domain of the Cell Wall Protein SraP Modulates Macrophage Apoptosis and Inflammatory Responses in *Staphylococcus aureus* Infections"

_molecules, 2025, doi:10.3390/molecules30051168_

Round 1
Reviewer 1 Report
Comments and Suggestions for Authors
The manuscript presents a well-structured and scientifically study on the role of the ligand-binding domain of the serine-rich adhesin for platelets (SraP) in Staphylococcus aureus pathogenesis, specially focusing on its effects on apoptosis, macrophage function, and bacterial survival. The study is well-supported by experimental data, but some points need to be improved
-The title is clear but need to be slightly more precise. e.g. The ligand binding domain of the cell wall protein SraP modulates macrophage apoptosis and inflammatory responses in Staphylococcus aureus infections."
-abstract should be should include a brief statement about study limitations or future directions.
-Some sentences are overly long and contain redundant information (e.g., Paragraph 3 discussing MRSA antibiotic resistance)
-in methods
-Clearly state how MW2 cultures were confirmed for purity and identity after genetic modifications.
-Describe how successful LLM deletion and overexpression were validated (e.g., sequencing, Western blot).
-in Macrophage Phagocytosis & Survival Assays, how were colony counts normalized between replicates? Were any statistical outliers removed?
-Specify primer sequences in a supplementary table, and how was RNA integrity verified (e.g., RIN score)?
-Figures should be more self-explanatory with improved legends, and the text should describe figure panels in a logical order (e.g., "Figure 1A shows... Figure 1B quantifies...").
-statistical comparisons for late apoptosis are missing. Were these differences non-significant
-discuss whether LLM-mediated effects are macrophage-specific or extend to other immune cells
-Some sentences are overly complex. Consider breaking them into shorter, more digestible statements.
Comments on the Quality of English Languageminor revision needed
Author Response
Responses to Reviewers’ comments
Reviewer comment #1
(x) The English could be improved to more clearly express the research.
RESPONSE: The reviewer’s comments and insightful suggestions are greatly appreciated. The manuscript has been proofread by a professional language editor from Springer Nature (see the attached certificate).
Comments and Suggestions for Authors
The manuscript presents a well-structured and scientifically study on the role of the ligand-binding domain of the serine-rich adhesin for platelets (SraP) in Staphylococcus aureus pathogenesis, specially focusing on its effects on apoptosis, macrophage function, and bacterial survival. The study is well-supported by experimental data, but some points need to be improved
-The title is clear but need to be slightly more precise. e.g. The ligand binding domain of the cell wall protein SraP modulates macrophage apoptosis and inflammatory responses in Staphylococcus aureus infections."
RESPONSE: We appreciate the reviewer’s constructive comments and suggestions. The title has changed as the reviewer recommended.
-abstract should be should include a brief statement about study limitations or future directions.
RESPONSE: A sentence on future directions has been added to the Abstract during the revision.
-Some sentences are overly long and contain redundant information (e.g., Paragraph 3 discussing MRSA antibiotic resistance)
RESPONSE: These sentences have been shortened and revised.
-in methods
-Clearly state how MW2 cultures were confirmed for purity and identity after genetic modifications.
RESPONSE: The information has been added to the Methods section. We used the Sanger sequencing to verify their purity and identity.
-Describe how successful LLM deletion and overexpression were validated (e.g., sequencing, Western blot).
RESPONSE: The information has been added to the Methods section. We have both sequencing and real-time PCR to validate the deletion and overexpression.
-in Macrophage Phagocytosis & Survival Assays, how were colony counts normalized between replicates? Were any statistical outliers removed?
RESPONSE: We followed a standard Gentamicin protection protocol for macrophage phagocytosis and survival assay (Su et al., 2012, Helminth infection impairs autophagy-mediated killing of bacterial enteropathogens by macrophages. J Immunol. 2012 Jun 25;189(3):1459–1466. doi: 10.4049/jimmunol.1200484) but also referenced another step-by-step protocol (Sharma and Puhar, 2019, Gentamicin Protection Assay to Determine the Number of Intracellular Bacteria during Infection of Human TC7 Intestinal Epithelial Cells by Shigella flexneri. Bio-protocol 9(13): e3292. DOI:10.21769/BioProtoc.3292). Basically, after cell lysis and serial dilution, 100 µl of the appropriate dilutions of the lysate were plated on LB agar plates. Four technical replicates were used per data/time point. After overnight incubation at 37°C, colonies on each plate were counted manually. The mean colony numbers from four plates were used to represent each time point. Because we only plated 100 µl of the dilution, the only normalization was to calculate mean colony number per ml for the final enumeration. No statistical outliers were removed.
-Specify primer sequences in a supplementary table, and how was RNA integrity verified (e.g., RIN score)?
RESPONSE: RNA integrity was verified using a BioAnalyzer 2000 (Agilent) with RIN number greater than 8.0. The primer sequences were listed in the supplementary files.
-Figures should be more self-explanatory with improved legends, and the text should describe figure panels in a logical order (e.g., "Figure 1A shows... Figure 1B quantifies...").
RESPONSE: Corrected.
-statistical comparisons for late apoptosis are missing. Were these differences non-significant
RESPONSE: No asterisk symbols in Figure 1C indicate that the no statistical significance
-discuss whether LLM-mediated effects are macrophage-specific or extend to other immune cells
RESPONSE: Great suggestions! Some sentences have been added during the revision (L304 – L311)
-Some sentences are overly complex. Consider breaking them into shorter, more digestible statements.
RESPONSE: Corrected as suggested.

Reviewer 2 Report
Comments and Suggestions for Authors
In the present work, the authors focused on the Staphylococcus aureus cell wall protein, serine-rich adhesin for platelets (SraP). The study provides experimental evidence that SraP influences macrophage functions in a human monocyte-derived macrophage model through its N-terminal L-lectin module (LLM) within the ligand-binding region. Flow cytometry was used to study that SraP serves a dual role in S. aureus pathogenesis—facilitating bacterial adhesion and invasion while also influencing macrophage function. The study of the LLM with inhibitors is important, it may offer a novel therapeutic approach for combating S. aureus infections.
It is important to mention that the work is a basic study providing an important information to the study of S. aureus. Without doubts, more techniques will complement the results. However, as an exploratory study I consider is enough and well focused.
The manuscript is well written and clear, the paper describes an interesting research work. The manuscript requires some minor revision before being published.
Minor details:
- There are few mistakes using italics in the discussion in the first 6 lines. It is necessary to correct it.
- There is a space in the line “However, the number of _survived cells after their engulf” that must be removed.
- Figure 3 and 4 must be reduced to look comparable to figure 2. They look oversized.
- As a suggestion, the sizes of the letters of the figures must be reduce, they are too big.
The work is suitable for publication in Molecules
Author Response
Reviewer comment #2
Comments and Suggestions for Authors
In the present work, the authors focused on the Staphylococcus aureus cell wall protein, serine-rich adhesin for platelets (SraP). The study provides experimental evidence that SraP influences macrophage functions in a human monocyte-derived macrophage model through its N-terminal L-lectin module (LLM) within the ligand-binding region. Flow cytometry was used to study that SraP serves a dual role in S. aureus pathogenesis—facilitating bacterial adhesion and invasion while also influencing macrophage function. The study of the LLM with inhibitors is important, it may offer a novel therapeutic approach for combating S. aureus infections.
RESPONSE: The authors appreciate the thorough reading of our manuscript by the reviewer. The comments are constructive and insightful. Indeed, we have just completed a follow-up study for systematic in silico screening for LLM inhibitors with promising results.
It is important to mention that the work is a basic study providing an important information to the study of S. aureus. Without doubts, more techniques will complement the results. However, as an exploratory study I consider is enough and well focused.
RESPONSE: Corrected.
The manuscript is well written and clear, the paper describes an interesting research work. The manuscript requires some minor revision before being published.
Minor details:
- There are few mistakes using italics in the discussion in the first 6 lines. It is necessary to correct it.
RESPONSE: Corrected.
- There is a space in the line “However, the number of _survived cells after their engulf” that must be removed.
RESPONSE: Corrected.
- Figure 3 and 4 must be reduced to look comparable to figure 2. They look oversized.
- As a suggestion, the sizes of the letters of the figures must be reduce, they are too big.
RESPONSE: Both figures were resized and reformatted during the revision.
The work is suitable for publication in Molecules

Round 2
Reviewer 1 Report
Comments and Suggestions for Authors
The manuscript was greatly improved and the authors respond to all comments
Comments on the Quality of English LanguageEnglish is clear